# A Self-Assembling Amphiphilic Peptide Dendrimer-Based Drug Delivery System for Cancer Therapy

**DOI:** 10.3390/pharmaceutics13071092

**Published:** 2021-07-17

**Authors:** Dandan Zhu, Huanle Zhang, Yuanzheng Huang, Baoping Lian, Chi Ma, Lili Han, Yu Chen, Shengmei Wu, Ning Li, Wenjie Zhang, Xiaoxuan Liu

**Affiliations:** 1State Key Laboratory of Natural Medicines and Jiangsu Key Laboratory of Drug Discovery for Metabolic Diseases, Center of Advanced Pharmaceuticals and Biomaterials, China Pharmaceutical University, Nanjing 210009, China; 1831070168@stu.cpu.edu.cn (D.Z.); 1621010179@stu.cpu.edu.cn (H.Z.); 3120070206@stu.cpu.edu.cn (Y.H.); 3119070191@stu.cpu.edu.cn (B.L.); 1731070155@stu.cpu.edu.cn (C.M.); 1821070945@stu.cpu.edu.cn (L.H.); 1821070944@stu.cpu.edu.cn (Y.C.); wenjiezhang@cpu.edu.cn (W.Z.); 2Department of Analytical Chemistry, College of Science, China Pharmaceutical University, Nanjing 210009, China; 1020081878@cpu.edu.cn; 3School of Pharmacy, Fujian Medical University, Fuzhou 350122, China

**Keywords:** amphiphilic peptide dendrimer, self-assembling, drug delivery, cancer therapy

## Abstract

Despite being a mainstay of clinical cancer treatment, chemotherapy is limited by its severe side effects and inherent or acquired drug resistance. Nanotechnology-based drug-delivery systems are widely expected to bring new hope for cancer therapy. These systems exploit the ability of nanomaterials to accumulate and deliver anticancer drugs at the tumor site via the enhanced permeability and retention effect. Here, we established a novel drug-delivery nanosystem based on amphiphilic peptide dendrimers (AmPDs) composed of a hydrophobic alkyl chain and a hydrophilic polylysine dendron with different generations (AmPD KK_2_ and AmPD KK_2_K_4_). These AmPDs assembled into nanoassemblies for efficient encapsulation of the anti-cancer drug doxorubicin (DOX). The AmPDs/DOX nanoformulations improved the intracellular uptake and accumulation of DOX in drug-resistant breast cancer cells and increased permeation in 3D multicellular tumor spheroids in comparison with free DOX. Thus, they exerted effective anticancer activity while circumventing drug resistance in 2D and 3D breast cancer models. Interestingly, AmPD KK_2_ bearing a smaller peptide dendron encapsulated DOX to form more stable nanoparticles than AmPD KK_2_K_4_ bearing a larger peptide dendron, resulting in better cellular uptake, penetration, and anti-proliferative activity. This may be because AmPD KK_2_ maintains a better balance between hydrophobicity and hydrophilicity to achieve optimal self-assembly, thereby facilitating more stable drug encapsulation and efficient drug release. Together, our study provides a promising perspective on the design of the safe and efficient cancer drug-delivery nanosystems based on the self-assembling amphiphilic peptide dendrimer.

## 1. Introduction

Cancer is one of the leading causes of death around the world [1]. Although considerable achievements have been made in clinical cancer treatment, an effective cure remains a challenge [2]. The efficacy of chemotherapy—the mainstay of clinical cancer treatment—is limited by its severe side effects, which include high toxicity, poor tumor selectivity, and inherent or acquired drug resistance during or after chemotherapy [3,4,5]. To overcome the side effects of chemotherapy, numerous therapeutic strategies have been proposed. One particularly promising strategy is nanotechnology-based drug delivery systems (NDDSs) [6,7,8,9]. These NDDSs are able to facilitate the accumulation and delivery of anticancer drugs at tumor lesions via the enhanced permeability and retention (EPR) effect by virtue of their unique nanoscale size. This can significantly increase the local concentration of the drugs and improve their therapeutic potency [10]. It is worth noting that the drug-loaded nanoparticles can be taken up by tumor cells via endocytosis, which can bypass drug efflux and increase the drug accumulation, hence overcoming drug resistance [11,12]. Therefore, the development of NDDSs brings new hope to revolutionize the therapeutic outcomes of cancer treatment.

Over the past decades, a variety of materials have been utilized to establish NDDSs for cancer therapy [7,13]. Among them, dendrimers—a special family of synthetic macromolecules furnished with a highly ramified architecture—have emerged as an attractive option because of their precisely defined structure and multivalent cooperativity [14,15]. In particular, amphiphilic dendrimers with judiciously tailored hydrophobic and hydrophilic components have been demonstrated to be able to self-assemble to supramolecular dendrimers for effective drug delivery in different disease models [16,17,18,19,20]. Recently, to combine the excellent properties of peptide dendrimers (such as their protein-like properties, good biocompatibility, etc. [21,22]), we developed amphiphilic peptide dendrimers (AmPDs), which carry peptide dendrons as hydrophilic heads for the delivery of nucleic acid therapeutics [23,24].

Herein, we exploit a novel NDDS based on AmPDs for the delivery of chemotherapeutics (Scheme 1). These AmPDs are composed of hydrophobic alkyl chain and hydrophilic polylysine dendron with different generations (AmPD KK_2_ and AmPD KK_2_K_4_). Based on the amphiphilic nature, these AmPDs would self-assemble to form supramolecular dendrimer nanoassemblies with hydrophobic cavities that can encapsulate hydrophobic chemotherapeutics. Doxorubicin (DOX) is used as the model chemotherapeutic to evaluate the drug delivery efficacy of AmPDs in a drug-resistant breast cancer model. The AmPD/DOX nanoparticles would be able to improve the intracellular uptake and accumulation of DOX in breast cancer cell lines, particularly drug-resistant breast cancer cells, therefore exerting effective anticancer activity while circumventing drug-resistance.

## 2. Materials and Methods

The full description of the materials and all the details of the related experiments are provided in the Appendix A.

### 2.1. Materials

Doxorubicin hydrochloride was purchased from Beijing Fengtai Hualian Co. Ltd. (Beijing, China). Cell Counting Kit-8 (CCK-8), 3-(4,5-dimethylthiazol-2-yl)-2,5-diphenyltetrazolium bromide (MTT) was purchased from Sigma-Aldrich (Merck Life Science, Shanghai, China).

Human MCF-7 breast cancer cells (doxorubicin-sensitive cell line) were purchased from the Tongpai Biotechnology Co. Ltd. (Shanghai, China). Human MCF-7R breast cancer cells (doxorubicin-resistant cell line) were provided by Prof. Hulin Jiang (China Pharmaceutical University, Nanjing, China).

All other reagents were from Energy Chemical Ltd. (Shanghai, China), Sinopharm Chemical Reagent Co, Ltd. (Shanghai, China), Aladdin (Shanghai, China) or Sigma Aldrich (Shanghai, China) and used without any further purification.

### 2.2. Synthesis of AmPD KK_2_

The synthetic protocol of hydrophobic alkyl chain and peptide dendrons was optimized (Supplementary Material) according to the reported literature [17,24,25]. AmPD 2–3, C_18_-N_3_, CuSO_4_·5H_2_O and NaAsc (l-Ascorbic Acid Sodium Salt) were dissolved in anhydrous THF under nitrogen atmosphere. Then the mixture was added to distilled water and stirred under nitrogen for 3 h at 50 °C. After solvent evaporation, the reaction mixture was extracted with CH_2_Cl_2_, washed with saturated NH_4_Cl solution, brine, and then dried over Na_2_SO_4_. The residue was purified by silica gel column chromatography, yielding AmPD KK_2_-Boc as white solid. Then AmPD KK_2_-Boc was dissolved in anhydrous CH_2_Cl_2_, and trifluoroacetic acid (TFA) was added to the above solution under stirring at 0 °C. The mixture was stirred under nitrogen for 4 h at room temperature. After solvent evaporation, the residue was washed with anhydrous diethyl ether. The product was further purified by dialysis using a dialysis tube, followed by lyophilization to give AmPD KK_2_ as white solid.

### 2.3. Synthesis of AmPD KK_2_K_4_

The synthetic protocol of AmPD KK_2_K_4_ was carried out similarly to the synthesis of AmPD KK_2_, yielding a white solid.

All the detailed synthetic processes and characterization data of AmPDs were in the Appendix A.

### 2.4. Critical Aggregation Concentration (CAC) of AmPDs Nanoassemblies

After sonicating for 5 min and resting at ambient temperature for 12 h, a fluorescence spectrophotometer was used to detect the AmPDs solution with Pyrene. The pyrene fluorescence spectra were recorded (an excitation wavelength: 334 nm).

### 2.5. Preparation of Doxorubicin-Loading Nanoformulations

The hydrophobic DOX was slowly added into the phosphate buffered saline (PBS) solution (0.01 M) containing AmPDs. The unencapsulated DOX was removed via a dialysis bag. The drug content loaded in the nanocarriers was calculated using the microplate reader (Cytation 5, BioTek, Winusky, VT, USA). The formulas of drug loading content and encapsulation efficiency are provided in the Appendix A.

### 2.6. Size Distribution, and Zeta Potential Measurements

The size distribution of AmPDs nanoassemblies and AmPDs/DOX nanoformulations was determined by dynamic light scattering (DLS) using NanoBrookOmni (Brookhaven, Long Island, NY, USA). The final concentrations of AmPDs in both the AmPDs nanoassemblies solution and AmPDs/DOX nanoformulations solution was 2.0 mg/mL.

### 2.7. In Vitro Drug Release

AmPDs/DOX nanoformulations were dissolved in the buffer (pH 7.4 or 5.0) and transferred into dialysis bags. Then, these dialysis bags were immersed into a relevant buffer and kept in a shaking bed. At a series of sequential time points, the amounts of released doxorubicin were detected using a microplate reader. The cumulative amount of DOX released from nanoparticles was plotted against time.

### 2.8. Cell Culture

MCF-7 cells were maintained in DMEM (HyClone™-GE, Logan, UT, USA), with 10% Foundation™ Fetal Bovine Serum (FBS) (Gemini Bio-Products, Riverside Parkway, West Sacramento, CA, USA) added. MCF-7R cells were maintained in RMPI 1640 (HyClone™-GE, Logan, UT, USA), supplemented with 10% FBS. MCF-7 and MCF-7R cells were incubated at 37 °C with 5% CO_2_.

### 2.9. In Vitro Anticancer Activity

The anticancer activities of AmPDs/DOX nanoformulations were performed on MCF-7 and MCF-7R cells. These cells were seeded and incubated at 37 °C for 24 h. After 24 h incubation, microculture tetrazolium solution was added and incubated. After removing the mediums, the cells were resuspended in dimethylsulfoxide (DMSO) solution. The absorbance of the DMSO solution was measured at 570 nm via a microplate reader. The cell metabolism toxicity and membrane damage toxicity of the blank carrier were also evaluated by MTT assay and lactate dehydrogenase (LDH) assay.

### 2.10. In Vitro Cellular Uptake

Flow cytometry: MCF-7R cells were seeded with a density of 6.0 × 10^4^ cells per well and cultured. Then, the culture mediums were replaced with free DOX and AmPDs/DOX nanoformulations. After 30 min and 2 h incubation, cells were digested, washed, and resuspended with PBS solution, then analyzed using flow cytometry.

Confocal microscopy: MCF-7R cells were seeded into confocal dishes. After 4 h of incubation, mediums containing samples (AmDPs/DOX nanoformulations or free DOX) were introduced into the system. After the removal of mediums, cells were washed, and stained with lysotracker green and Hoechst 33,342. The cellular uptake of nanoformulations and free DOX were observed through two-photon confocal microscope (Zeiss, Oberkochen, Germany).

### 2.11. Drug Penetration in 3D-Cultured Tumor Spheroids

The MCF-7R 3D-cultured tumor spheroids were incubated with the free DOX or AmPDs/DOX nanoformulations. Four hours later, the medium containing the free DOX or AmPDs/DOX nanoformulations were removed and the tumor spheroids were washed and transferred to confocal dishes. The penetration of the 3D-cultured tumor spheroids at different depths was observed by a two-photon confocal microscope.

### 2.12. In Vitro Anticancer Activity in 3D-Cultured Tumor Spheroids

The MCF-7R 3D-cultured tumor spheroids were treated with culture mediums including AmPDs/DOX nanoformulations and free DOX at a serial of doxorubicin concentrations for 48 h. After adding the CCK-8 working solution into each well, the 3D-cultured tumor spheroids were incubated at 37 °C for 4 h. Then, the absorbance was measured via the microplate reader.

### 2.13. Statistical Tests

All data are presented as mean ± SD unless otherwise indicated. Statistical analysis was performed by one-way ANOVA or two-way ANOVA with Tukey’s post-hoc test (Graphpad Prism 8.01). * *p* ≤ 0.05, ** *p* ≤ 0.01, and *** *p* ≤ 0.001.

## 3. Results and Discussion

### 3.1. Synthesis of the Amphiphilic Peptide Dendrimers (AmPDs)

AmPDs composed of hydrophobic C_18_ alkyl chain and different hydrophilic poly(L-lysine) peptide dendrons (AmPD KK_2_ and AmPD KK_2_K_4_) were synthesized according to the strategy described in Appendix A). Different generations of Boc groups protected hydrophilic poly(L-lysine) dendrons bearing alkyne groups, and the hydrophobic C_18_ alkyl chains bearing an azide group were prepared using a previously reported protocol [17,24,25]. Then, hydrophilic dendrons were covalently conjugated with the hydrophobic alkyl chains via robust and efficient Cu(I)-catalyzed azide–alkyne cycloaddition (CuAAC) ‘click’ reaction to yield AmPDs with protecting groups. After the removal of their protecting groups (Boc groups), the terminal amines were exposed to obtain the AmPDs. The synthesis is described in further detail in the Supporting Information. The structures of these AmPDs were characterized using ^1^H NMR and mass spectrometry (Appendix A).

### 3.2. Characterization of Self-Assembly Behaviours of AmPDs

Because of their amphiphilic nature, the AmPDs can self-assemble into nanoassemblies in an aqueous environment. The critical aggregation concentration (CAC) of AmPD KK_2_K_4_ (14 μM) was 2.3 times that of AmPD KK_2_ (6.1 μM) (Figure 1A,B), indicating that AmPD KK_2_ is more inclined to self-assemble into nanoassemblies than AmPD KK_2_K_4_. This might be because AmPD KK_2_ possesses a more favorable balance between its hydrophilic dendron and hydrophobic chain. Dynamic light scattering analysis revealed that the hydrodynamic sizes of the AmPD KK_2_ and AmPD KK_2_K_4_ nanoassemblies were approximately 9.4 and 15 nm, respectively. Moreover, the zeta potential of AmPD KK_2_ was 12.3 mV, which was slightly higher than that of AmPD KK_2_K_4_ (8.50 mV) (Figure 1C and Appendix A). We also examined the secondary configurations of AmPD nanoassemblies using circular dichroism. The results shown in Figure 1D indicate that the AmPD KK_2_ and AmPD KK_2_K_4_ nanoassemblies had similar secondary structures. These similarities were also confirmed by data analysis using CDNN software (Appendix A). These results demonstrated that the AmPD nanoassemblies retain the inherent properties of polylysine.

### 3.3. Drug Encapsulation and Drug Release Profiles of DOX-Loaded AmPD Nanoformulations

DOX is a widely used, broad-spectrum anticancer drug that functions by intercalating into DNA to inhibit nucleic-acid synthesis. We selected DOX as a model drug to investigate drug encapsulation by the AmPD nanoassemblies. We used the film-dispersion method to prepare two DOX-loaded AmPD formulations: AmPD KK_2_/DOX and AmPD KK_2_K_4_/DOX. These two formulations had a similar drug-loading content (~19%) and encapsulation efficiency (~97%) (Figure 2A). The size distribution of the AmPD KK_2_/DOX and AmPD KK_2_K_4_/DOX formulations was approximately 73 and 80 nm, respectively (Figure 2B and Appendix A). Their surface zeta potentials were 13.4 and 11.6 mV, respectively, indicating that they were in a stable colloidal state. These results demonstrated that the AmPD nanoassamblies could effectively package the hydrophobic anti-tumor drugs (DOX) via hydrophobic interaction to form stable DOX-loaded AmPDs nanoassemblies.

Controllable release of the loaded therapeutics at the tumor site is an important property of an effective drug delivery system. The acidity of the tumor microenvironment is lower than the normal tissue [26,27,28]. Thus, the ideal DDS should be able to reduce the release of the loaded drugs as little as possible under physiological conditions (pH 7.4) for safety consideration, while promoting drug release as much as possible under the acidic condition (pH 5.0) at the tumor site for therapeutic purposes. Hence, we evaluated the drug release profile of AmPDs/DOX nanoformulations at different pH values (5.0 and 7.4). The results showed that DOX was rapidly and efficiently released from the AmPDs/DOX nanoformulations at pH 5.0, with a cumulative release of more than 50% within 24 h (Figure 2C). The drug-release behavior of the AmPD KK_2_–DOX and AmPD KK_2_K_4_–DOX nanoformulations was similar. We attribute this to the protonation of the encapsulated amine-bearing DOX at pH 5.0, resulting in electrostatic repulsion with positively charged amine-containing AmPD nanoassemblies, which promoted drug release under acidic conditions. However, in pH 7.4 buffer, the amount of drug released from AmPD KK_2_/DOX (about 25%) was substantially less than that from AmPD KK_2_K_4_/DOX (about 41%). This difference is probably due to the better self-assembly capacity of AmPD KK_2_, which forms more stable formulations with hydrophobic drugs than AmPD KK_2_K_4_, thereby providing better protection of the loaded cargo from leakage under physiological pH.

### 3.4. Potent Anticancer Efficacy of AmPD/DOX Nanoformulations via Effective Intracellular Uptake

After evaluating the drug-release characteristics of the AmPD/DOX nanoformulations, we evaluated their anticancer efficacy in human breast cancer cell lines, including DOX-sensitive MCF-7 cells and DOX resistant MCF-7R cells. First, we used MTT assays to examine their antiproliferative performance. In the DOX-sensitive MCF-7 cells, DOX-loaded AmPD nanoassemblies efficiently inhibited cell proliferation (Appendix A); the half-maximal inhibitory concentrations (IC_50_) were 2.4 and 2.6 μg/mL (or 4.4 and 4.9 μM) for AmPD KK_2_/DOX and AmPD KK_2_K_4_/DOX, respectively; these values were similar to that of free DOX (Appendix A). By contrast, in the DOX-resistant MCF-7R cells, AmPD/DOX had a much better anticancer effect with free DOX (Figure 3A and Appendix A). Interestingly, the IC_50_ of AmPD KK_2_K_4_/DOX (7.0 μg/mL or 47.8 μM) was approximately 2.7 times greater than that of AmPD KK_2_/DOX (26 μg/mL or 12.9 μM) (Appendix A), indicating that AmPD KK_2_/DOX induced a more potent antiproliferative effect than AmPD KK_2_K_4_/DOX in MCF-7R cells.

We hypothesize that the different antiproliferative effects of the two AmPD/DOX nanoformulations in the drug-resistant cell line may be due to differences in their intracellular uptake. To validate this hypothesis, we carried out flow cytometry to quantify the intracellular uptake. As shown in Figure 3B, AmPDs/DOX nanoformulations facilitated efficient intracellular uptake of DOX in MCF-7R cells in a time- and dose-dependent manner. AmPD KK_2_/DOX exhibited more effective cellular uptake of DOX than AmPD KK_2_K_4_/DOX at all times points and dosages. Such enhanced cellular uptake of AmPDs/DOX nanoformulations in MCF-7R cells was further confirmed by using confocal laser scanning microscopy (CLSM) (Appendix A). Moreover, after an additional 8 h of incubation, stronger fluorescent signals of DOX were detected for treatment with AmPD KK_2_/DOX than with AmPD KK_2_K_4_/DOX (Appendix A), indicating more efficient intracellular accumulation of AmPD KK_2_/DOX in MCF-7R cells.

We then assessed the safety profile of the AmPD delivery system in the two cell lines. As we expected, no notable metabolite toxicity was found even at a high concentration of AmPDs using MTT assays (Figure 3C and Appendix A), and no obvious damage to the cell membrane was detected by LDH assays (Figure 3D and Appendix A). This confirms the non-toxic characteristics of the AmPD delivery system.

Collectively, these results suggested that AmPD-based nanoassemblies can successfully deliver DOX into drug-resistant MCF-7R cancer cells, enhance the intracellular retention of DOX, and thereby induce a potent anticancer effect. Interestingly, AmPD KK_2_ facilitated more efficient intracellular uptake and retention of DOX than AmPD KK_2_K_4_, thereby more effectively inhibiting cell proliferation.

### 3.5. Deep Drug Penetration and Cellular Uptake in 3D-Cultured Tumor Spheroids

3D tumor spheroids can retain the material and structural basis of the tumor microenvironment, rendering them an attractive in vitro model that mimics the real tumor environment [29,30]. Thus, we used employed 3D-cultured tumor spheroids to study the drug delivery mediated by AmPDs. First, we utilized CLSM measurements to trace the penetration and uptake behavior of the AmPDs/DOX nanoformulations in MCF-7R tumor spheroids. Strong fluorescent signals of DOX were observed for treatments with the AmPDs/DOX nanoformulations, whereas very weak signals were observed upon treatment with free DOX (Figure 4A,B). These results unambiguously confirm that, in contrast with free DOX, the AmPDs/DOX nanoformulations penetrated deep into the interior of the tumor spheroids.

We further measured the cellular uptake of the AmPDs/DOX nanoformulations in MCF-7R cells inside the tumor spheroids using flow cytometry. As illustrated in Figure 4C, AmPDs/DOX nanoformulations substantially enhanced the uptake and accumulation of DOX in MCF-7R cells of the tumor spheroids, which corroborates the results of the CLSM measurements. Similar to the situation in the 2D model, AmPD KK_2_/DOX exhibited better penetration behavior than AmPD KK_2_K_4_/DOX in the 3D tumor spheroids. This difference in performance might be due to the more stable AmPD KK_2_/DOX formulation better protecting the DOX from leakage before penetrating into the interior of the tumor spheroid. This would result in more efficient penetration and cellular uptake of DOX in the tumor cells inside the spheroids.

### 3.6. Enhanced Antiproliferative Effect in 3D-Cultured Tumor Spheroids

Encouraged by the enhanced penetration and cellular internalization of the AmPDs/DOX nanoformulations in the tumor spheroids, we further evaluated their antiproliferative effect. The results shown in Figure 4D suggest that the AmPDs/DOX nanoformulations induce a potent, dose-dependent antiproliferative effect, in contrast with free DOX, which did not inhibit cell proliferation. Specifically, the IC_50_ of AmPD KK_2_/DOX (48.4 μg/mL or 89.1 μM) was approximately 3.5 times lower than AmPD KK_2_K_4_/DOX (138.5 μg/mL or 254.8 μM) (Appendix A), showing that proliferation was much more efficiently inhibited by treatment with AmPD KK_2_/DOX. We attribute this enhanced antiproliferative effect to the more efficient penetration and internalization of AmPD KK_2_/DOX in the tumor spheroids. These results demonstrate that although both AmPDs/DOX nanoformulations can effectively inhibit the proliferation of 3D tumor spheroids, AmPD KK_2_/DOX is the better potential candidate for cancer therapy.

## 4. Conclusions

Despite being a mainstay in clinical treatment, chemotherapy is limited by its severe side effects and inherent or acquired drug resistance. Nanotechnology-based drug-delivery systems are widely expected to improve therapeutic efficacy while reducing toxicity for anticancer treatment. In this study, we developed a novel self-assembling drug-delivery system based on amphiphilic peptide dendrimers (AmPDs) bearing a hydrophobic C_18_ chain and a hydrophilic peptide dendron with different generations (AmPD KK_2_ and AmPD KK_2_K_4_). The AmPDs self-assembled into nanoassemblies and effectively encapsulate the antitumor drug (DOX) to form stable nanoformulations (AmPD KK_2_/DOX and AmPD KK_2_K_4_/DOX). The AmPD/DOX nanoformulations conquered drug resistance in drug-resistant breast cancer MCF-7R cells owing to their enhanced intracellular uptake and accumulation of DOX, and in 3D-cultured tumor spheroids owing to their efficient penetration. Thus, a potent anticancer effect was achieved in 2D and 3D breast cancer models. Interestingly, AmPD KK_2_, which had a smaller peptide dendron than AmPD KK_2_K_4_, can encapsulate DOX to form more stable nanoparticles, resulting in better cellular uptake, penetration, and anti-proliferative activity. This may be because AmPD KK_2_ maintains a better balance between hydrophobic and hydrophilic entities. Collectively, this work provides a promising new perspective on the design of safe and efficient drug-delivery platforms for cancer therapy based on self-assembling amphiphilic peptide dendrimers.

## Data Availability

Not applicable.

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
