# Peer review of "A Self-Assembling Amphiphilic Peptide Dendrimer-Based Drug Delivery System for Cancer Therapy"

_pharmaceutics, 2021, doi:10.3390/pharmaceutics13071092_

Round 1

Reviewer 1 Report

The paper deals with DDS built of amphipilic dendrons obtained from polylysine with attached hydrocarbon moiety by click-chemistry protocol. Two various size polyK dendrons were applied, namely KK2 and KK2K4, and derivitized with propargylamine, followed by condensation with octadecane azide to obtain two various size AmPDs. The description of chemistry is very precise and reproducible (Supplementary Materials).

Obtained AmPDs spontaneously accumulate into spheroidal supramolecular objects of the size determined by DLS. They also are able to encapsulate DOX up to 19 weight %. Encapsulation of DOX leads to distinct enlargement of nanoparticle size from 9.4 to 73 nm for AmPDKK2 and from 15 to 79 for for AmPDKK2K4. Finally the DOX@AmPD formulation were tested on MCF-7 and DOX-resistant MCF-7R cell lines as DDS. It was demonstrated that IC-50 determined for MCF was slightly lower for both DDS than for DOX itself. It seems that this result is not striking.

However, the story begins afterwards: first MCF-7R were sensitive to DOX delivered by AmPDKK2, with reasonable IC50. Further the accumulation of DOX was evidenced in 3D-cultured tumor spheroidal MCF-7R by CM which in combination with antiproliferative effect induced by DOX@AmPDKK2 is THE results. Although part of paper seems regular work as many others, the results obtained on 3-D cultured tumor is added value.

Everything is done very precisely, nicely written and clear.

As a reviewer I was looking also for some small mistakes, which can be easily corrected by authors. Here are my remarks:

1.It would be nice to add the micromolar concentration of DOX at least for IC50 (Table S4 and Main text, section 3.4)

2.Line 110; might be better is: ambient instead of interior temperature

3.Line 199 is:… inherent properties of natural proteins… Maybe better to write properties of polylysine (which is rather not natural protein)

SUPPLEMENTARY MATERIALS

Page 4 and page 5: Some abbreviations like DMF, HBTU, DIPEA, HOBT, TFA, DCM are widely used. I could not identify what is PE/EA (eluent for silica chrom). If DCM is used, please replace CH2Cl2 with DCM (page 6)

Page 5: “was added to the above solution …” is used twice in two consecutive sentences, please do something with this.

Page 7: Instead of: then the mixture was added distilled water….. Replace with: “Distilled water (1.00 mL) was added to the mixture and stirred ….

Page 8: Doxorubicin hydrochloride was dissolved in 1.0 mL mixing solvent, adding trimethylamine to obtain hydrophobic DOX. I am not sure, but suggest: DOX HCl was dissolved in CHCl3/MeOH = 1/3 and TEA was added (amount) to it.

Page 9: Some of authors just wrote the protocol: Add 100 microL of LDH working solution to each well, incubate….

Probably it is better to write what has been done: 100 microL of LDH working solution was added to each well….. and so on.

Dear Authors, I hope I was helpful.

Paper is excellent, I wish I have such methods in hand, you are happy people. Best wishes for your research and good health for all of you in these dangerous times

Author Response

The paper deals with DDS built of amphipilic dendrons obtained from polylysine with attached hydrocarbon moiety by click-chemistry protocol. Two various size polyK dendrons were applied, namely KK2 and KK2K4, and derivitized with propargylamine, followed by condensation with octadecane azide to obtain two various size AmPDs. The description of chemistry is very precise and reproducible (Supplementary Materials).

Obtained AmPDs spontaneously accumulate into spheroidal supramolecular objects of the size determined by DLS. They also are able to encapsulate DOX up to 19 weight %. Encapsulation of DOX leads to distinct enlargement of nanoparticle size from 9.4 to 73 nm for AmPD KK2 and from 15 to 79 for for AmPD KK2K4. Finally the DOX@AmPD formulation were tested on MCF-7 and DOX-resistant MCF-7R cell lines as DDS. It was demonstrated that IC50 determined for MCF was slightly lower for both DDS than for DOX itself. It seems that this result is not striking.

However, the story begins afterwards: first MCF-7R were sensitive to DOX delivered by AmPD KK2, with reasonable IC50. Further the accumulation of DOX was evidenced in 3D-cultured tumor spheroidal MCF-7R by CM which in combination with antiproliferative effect induced by DOX@AmPD KK2 is THE results. Although part of paper seems regular work as many others, the results obtained on 3-D cultured tumor is added value.

Everything is done very precisely, nicely written and clear.

As a reviewer I was looking also for some small mistakes, which can be easily corrected by authors. Here are my remarks:

1、It would be nice to add the micromolar concentration of DOX at least for IC50 (Table S4 and Main text, section 3.4)

A1. Thanks for the reviewer’s kind suggestion. Since we used mass concentration (µg/mL) for all the figures and tables in the main text and the supplementary materials, we therefore used mass concentration for IC50 value too. According to the suggestion of the reviewer, we have added micromolar concentration in section 3.4 of the revised manuscript and Table S4.

2、Line 110; might be better is: ambient instead of interior temperature

A2. We sincerely apologize for this mistake. We have revised it accordingly in the revised manuscript.

3、Line 199 is:… inherent properties of natural proteins… Maybe better to write properties of polylysine (which is rather not natural protein)

A3. Thanks for the reviewer’s kind suggestion. We have modified it in the revised manuscript and highlighted it in green.

SUPPLEMENTARY MATERIALS

Page 4 and page 5: Some abbreviations like DMF, HBTU, DIPEA, HOBT, TFA, DCM are widely used. I could not identify what is PE/EA (eluent for silica chrom). If DCM is used, please replace CH2Cl2 with DCM (page 6)

A4. We sincerely apologize for these mistakes. We have modified the “PE/EA (1/1)” on page 4 of the supplementary materials to “Petroleum ether (PE) / ethyl acetate (EA) (1/1)”, and replace CH2Cl2 with DCM on page 6 and 7. The modified content is highlighted in the revised supplementary materials.

Page 5: “was added to the above solution …” is used twice in two consecutive sentences, please do something with this.

A5. Thanks for the reviewer’s comments. We have revised this sentence as follows:

AmPD 2-3: Boc-L-Lys(Boc)-OH (566 mg, 1.63 mmol), HBTU (620 mg, 1.63 mmol), HOBT (221 mg, 1.63 mmol) and DIPEA (1.20 mL, 6.81 mmol) were dissolved in anhydrous DMF (3.00 mL) under nitrogen atmosphere and stirred at 0oC for 30 min. A solution of AmPD 2-2 (280 mg, 0.68 mmol) in DMF (2.00 mL) was added to the above solution under stirring. The reaction solution was moved to 30oC and continued to stir under nitrogen for 48 h.

Page 7: Instead of: then the mixture was added distilled water….. Replace with: “Distilled water (1.00 mL) was added to the mixture and stirred ….

A6. Thanks for the reviewer’s kind suggestion. We have modified it in the revised supplementary materials and highlighted it in green.

Page 8: Doxorubicin hydrochloride was dissolved in 1.0 mL mixing solvent, adding trimethylamine to obtain hydrophobic DOX. I am not sure, but suggest: DOX HCl was dissolved in CHCl3/MeOH = 1/3 and TEA was added (amount) to it.

A7. We thank the reviewer for the constructive comments. We have modified it in the revised supplementary materials and highlighted it in green.

Page 9: Some of authors just wrote the protocol: Add 100 microL of LDH working solution to each well, incubate….

Probably it is better to write what has been done: 100 microL of LDH working solution was added to each well….. and so on.

A8. Thanks for the reviewer’s kind suggestion. We have added the description of the protocol in the revised supplementary materials as follows:

100 μL of LDH working solution was added to each well, incubate for another 20 minutes, then add 50 μL of stop solution was added to each well, and immediately measure the absorbance at 490 nm.

Dear Authors, I hope I was helpful.

Paper is excellent, I wish I have such methods in hand, you are happy people. Best wishes for your research and good health for all of you in these dangerous times

A9. Thanks so much for the kind wishes from the reviewer. We also hope that everything goes well on their side.

Reviewer 2 Report

The article describes the synthesis of new peptide dendrimers and their use for transport of doxorubicin into cancer cells. The topic is new and promising. The article is very well written and easy to read and understand. The authors used the modern techniques to support the tasks. The results are correctly done and well described. The article can be published after small improvements.

I have only minor remarks:

1) discussion is too short. Please, add some references on similar studies and briefly describe the difference between your results and other results.

2) Please, add the conclusions. For example, the present Discussion section can be transferred into Conclusions section. But Discussion section is practically absent at present time.

Author Response

Thanks for the reviewer’s kind suggestion. In fact, when we prepared the manuscript, we put our discussion part together with the results part. However, when we transferred the manuscript into the template of this journal, we missed the discussion part. We are very sorry for that. We have modified the title of the third part to "Results and Discussion" and the title of the fourth part to "Conclusion" in the revision version and highlighted them in green.

Reviewer 3 Report

In their manuscript, Dandan Zhu et al. presented the synthesis and characterization of DOX loaded into peptide dendrimers as drug delivery system for cancer therapy. Globally, the manuscript is well-written and the experimental part is well-detailed in the Supporting Information part. However, despites the interest of the obtained results, some data are missing thus leading to too fast conclusions.

In view of these general remarks and the following ones, I do recommend the publication of the manuscript in Pharmaceutics after major revision.

Please find below specific comments and questions that need to be addressed before any publication.

  1. Page 2, Scheme 1: The authors have to mention to what the blue and yellow parts of the amphiphilic peptide dendrimers correspond to.
  2. Page 3, Line 94: The authors have to give the meaning of the abbreviation “NaAsc”.
  3. Page 4, lines 122-123 and page 8 SI n°5: The sentence “The final concentrations of … were both 2.0 mg/mL.” needs to be reworded.
  4. Page 4, line 146 and Page 10 SI n°10: The word “planted” has to be changed to “seeded”.
  5. Page 6, lines 207-208: Why did the authors selected the DOX as drug model?
  6. Page 6, line 211: How can the authors explain the significant increase of their nanoformulation sizes after DOX encapsulation (9 and 15 nm for empty nanoformulations to 73 and 80 nm for DOX-loaded nanoformulations?
  7. Page 6, lines 213-214: Normally, it is commonly admitted that nanovectors with an absolute value of zeta potential equal or higher than 30 mV are electrostatically stabilized (1. Patel, V.R.; Agrawal, Y.K. Nanosuspension: an approach to enhance solubility of drugs. J. Adv. Pharm. Technol. Res. 2011, 2, 81-87. 2. Pearce, A.K.; O’Reilly, R.K. Insight into active targeting of nanoparticles in drug delivery: Advances in clinical studies and design considerations for cancer nanomedicine. Bioconjugate Chem. 2019, 30, 2300-2311. 3. Elsabahy, M.; Wooley, K.L. Design of polymeric nanoparticles for biomedical delivery applications. Chem. Soc. Rev. 2012, 41, 2545–2561.)
  8. Page 6, lines 230-234: “The results showed that DOX … within 24 hours (Fig. 2C)”, I don’t totally agree with this conclusion since the difference in drug release at pH5 and pH7.4 is not so important, and a significant DOX release was at pH7.4.
  9. Page 7, line 245: “After confirming the acid-promoted drug-release characteristics”, It is not so obvious. The authors have to be more moderated.
  10. Page 8, lines 274-283: Do the authors study the mechanism of cell up-take? Is the DOX-loaded nanovectors penetrate into the cells via endocytosis?
  11. Page 9, “Discussion”: In my opinion, it is not a discussion part but rather a conclusion part.
  12. Page 21 SI, Table S2: The authors have to change the words “Heilx” to “Helix” and “Rndm. Coil” to “Random Coil”.

Author Response

In their manuscript, Dandan Zhu et al. presented the synthesis and characterization of DOX loaded into peptide dendrimers as drug delivery system for cancer therapy. Globally, the manuscript is well-written and the experimental part is well-detailed in the Supporting Information part. However, despites the interest of the obtained results, some data are missing thus leading to too fast conclusions.

In view of these general remarks and the following ones, I do recommend the publication of the manuscript in Pharmaceutics after major revision.

Please find below specific comments and questions that need to be addressed before any publication.

1、Page 2, Scheme 1: The authors have to mention to what the blue and yellow parts of the amphiphilic peptide dendrimers correspond to.

A1. Thanks for the reviewer’s kind suggestion. The yellow parts of the amphiphilic peptide dendrimers represent the hydrophilic polylysine dendrons, and the blue parts represent the alkyl chain. We have added this description of the different colors in the structure of AmPDs in Scheme 1 and highlighted it in green in the revised manuscript.

2、Page 3, Line 94: The authors have to give the meaning of the abbreviation “NaAsc”.

A2. Thanks for the reviewer’s kind suggestion. We have added the introduction of “NaAsc” in the revised manuscript.

3、Page 4, lines 122-123 and page 8 SI n5: The sentence “The final concentrations of … were both 2.0 mg/mL.” needs to be reworded.

A3. We sincerely apologize for the mistakes. We have revised this sentence as following and highlighted them in green in the revised version:

The final concentrations of AmPDs in both the AmPDs nanoassemblies solution and AmPDs/DOX nanoformulations solution was 2.0 mg/mL.

4、Page 4, line 146 and Page 10 SI n10: The word “planted” has to be changed to “seeded”.

A4. Thanks for the reviewer’s kind suggestion. We have modified them in the revised manuscript and highlighted them in green.

5、Page 6, lines 207-208: Why did the authors selected the DOX as drug model?

A5. We do appreciate the reviewer’s question. DOX is a widely used, broad-spectrum anticancer drug that functions by intercalating into DNA to inhibit nucleic acid synthesis. However, high systemic toxicity and drug resistance always accompany DOX, leading to a high recurrence rate and therapeutic failure in cancer treatment. Therefore, we chose DOX as drug model for a proof-of-concept study to verify the safety and efficacy of the self-assembling amphiphilic peptide dendrimer-based drug delivery platform. We expected that our dendrimer-based drug delivery platform can mediate safe and efficient delivery of anticancer drug DOX to exert potent anti-tumor effects.

6、Page 6, line 211: How can the authors explain the significant increase of their nanoformulation sizes after DOX encapsulation (9 and 15 nm for empty nanoformulations to 73 and 80 nm for DOX-loaded nanoformulations?

A6. We do appreciate the reviewer’s comment. The AmPDs/DOX nanoformulations were indeed larger in size than the empty AmPDs nanoassemblies. The AmPDs bearing hydrophobic alkyl chains and hydrophilic polylysine dendrons could interact with DOX via hydrophobic interaction. The increase in size could be reasonably ascribed to the expansion in assemblies dimensions upon accommodating DOX within the hydrophobic core to reach high loading. In addition, the DOX-loaded AmPPDs could further aggregat to form nanoassemblies with secondary structures by virtue of the inherent properties of polylysine. All this together may explain the significant increase in size of AmPPDs based nanoformulation after DOX encapsulation.

7、Page 6, lines 213-214: Normally, it is commonly admitted that nanovectors with an absolute value of zeta potential equal or higher than 30 mV are electrostatically stabilized (1.Patel, V.R.; Agrawal, Y.K. Nanosuspension: an approach to enhance solubility of drugs.J. Adv. Pharm. Technol. Res.2011,2, 81-87.2.Pearce, A.K.; O’Reilly, R.K. Insight into active targeting of nanoparticles in drug delivery: Advances in clinical studies and design considerations for cancer nanomedicine. Bioconjugate Chem.2019,30, 2300-2311.3.Elsabahy, M.; Wooley, K.L. Design of polymeric nanoparticles for biomedical delivery applications. Chem. Soc. Rev.2012,41, 2545–2561.)

A7. We do appreciate the reviewer’s comment. After carefully checking the literatures, we revised the description of Zeta potential according to the suggestions from the reviewer and highlighted in green in the revised manuscript.

8、Page 6, lines 230-234: “The results showed that DOX … within 24 hours (Fig. 2C)”, I don’t totally agree with this conclusion since the difference in drug release at pH 5 and pH 7.4 is not so important, and a significant DOX release was at pH 7.4.

A8. We do appreciate the reviewer’s comment. As we mentioned in the manuscript, we expected our dendrimer-based drug delivery system should be able to reduce the release of the loaded drugs as less as possible under physiological condition (pH 7.4) for safety consideration, while promoting drug release as much as possible under the acidic condition (pH 5.0) at tumor site for therapeutic purpose. We thus evaluated drug release profiles of DOX-loaded AmPDs in different pH media. At pH 5.0 buffers, the release rate of DOX from DOX-loaded AmPD KK2 could reach to around 60% within 24 hours, which was significantly better than the release rate at pH 7.4 (around 20%). Therefore, we believe this pH-dependent drug release behavior is beneficial for drug delivery in the tumor site.

As reviewer mentioned, we indeed observed around 20% drug release rate at pH 7.4 with one of our AmPD-based delivery systems. Such drug release rate at pH 7.4 is more or less similar to the reported results obtained with amphiphilic peptide-based drug delivery systems in some recent literatures (Colloids and Surfaces B: Biointerfaces 102 (2013) 833– 841, Colloids and Surfaces B: Biointerfaces 114 (2014) 398– 403, Mater. Res. Express 6 (2019) 115411, and Biomater. Sci., 2018, 6, 774). Anyway, we totally agree with the reviewer that the release rate at pH 7.4 need to be strictly controlled and the release rate at pH 5.0 need to be further improved with the aim to develop idea drug delivery systems. Therefore, we are actively working on this direction and wish to report more promising results in the near future.

9、Page 7, line 245: “After confirming the acid-promoted drug-release characteristics”, It is not so obvious. The authors have to be more moderated.

A9. We do appreciate the reviewer’s suggestion. We have modified it as following and highlighted it in green in the revised manuscript:

After evaluating the drug-release characteristics of the AmPDs/DOX nanoformulations

10、Page 8, lines 274-283: Do the authors study the mechanism of cell up-take? Is the DOX-loaded nanovectors penetrate into the cells via endocytosis?

A10. We do appreciate the reviewer’s suggestion. We indeed evaluated the cellular uptake mechanism of the DOX-loaded AmPPD nanovectors. We observed that the fluorescence intensity of the DOX-loaded AmPPD nanovectors in the MCF-7R cells was significantly suppressed after incubating with the endocytic inhibitor-cytochalasin D (the inhibitor of macropinocytosis), while no significant suppression was observed in presence of other two inhibitors-genistein (the inhibitor of caveolae-mediated endocytosis) and chlorpromazine (the inhibitor of clathrin-mediated endocytosis) (Figure below). These results suggested that DOX-loaded AmPDs nanoparticles enter into tumor cells mainly through macropinocytosis-mediated endocytosis. Since we did not observe significant difference in the cellular uptake mechanism of AmPD KK2/DOX and AmPD KK2K4/DOX in MCF-7R cells, we therefore didn’t present and discuss the cellular uptake mechanism of the DOX-loaded nanovectors in our submitted manuscript.

Figure: The cellular uptake mechanism of AmPD KK2/DOX (A) and AmPD KK2K4/DOX in MCF-7R cells in the presence of different inhibitors. The inhibitors are Cytochalasin D for macropinocytosis, chlorpromazine for clathrin-mediated endocytosis and genistein for caveolae-mediated endocytosis. (mean ± SD, n = 3, ***p < 0.001 represented Cytochalasin D vs Control).

11、Page 9, “Discussion”: In my opinion, it is not a discussion part but rather a conclusion part.

A11. We sincerely appreciate the reviewer’s comments. In fact, when we prepared the manuscript, we put our discussion part together with the results part. However, when we transferred the manuscript into the template of this journal, we missed the discussion part. We are very sorry for that. We have modified the title of the third part to "Results and Discussion" and the title of the fourth part to "Conclusion" in the revision version and highlighted them in green.

12、Page 21 SI, Table S2: The authors have to change the words “Heilx” to “Helix” and “Rndm. Coil” to “Random Coil”.

A12. We sincerely apologize for these mistakes. We have corrected them in the revised version.

Round 2

Reviewer 1 Report

This paper was already very good. After introducing language corrections is almost perfect. Some grammar corrections in Supplement can be introduced on Editorail level (look at Section 8; Suppl.)

I recommned this work to publish

Reviewer 3 Report

The authors have correctly answered to all the questions raised by the reviewers and significantly improved their manuscript.

I therefore recommend this manuscript for publication in Pharmaceutics.